# Invasive Crayfish *Faxonius limosus*: Meat Safety, Nutritional Quality and Sensory Profile

**DOI:** 10.3390/ijerph192416819

**Published:** 2022-12-14

**Authors:** Jasmina Lazarević, Ivana Čabarkapa, Slađana Rakita, Maja Banjac, Zorica Tomičić, Dubravka Škrobot, Goran Radivojević, Bojana Kalenjuk Pivarski, Dragan Tešanović

**Affiliations:** 1Institute of Food Technology, University of Novi Sad, 21000 Novi Sad, Serbia; 2Department of Geography, Tourism and Hotel Management, Faculty of Sciences, University of Novi Sad, 21000 Novi Sad, Serbia

**Keywords:** spiny-cheek invasive crayfish *Faxonius limosus*, meat safety, nutritional quality, sensory profile

## Abstract

The aim of the present study was to evaluate the safety parameters, nutritional value and sensory profile of the meat from spiny-cheek crayfish (*Faxonius limosus*), captured from the Danube River in Serbia. To achieve this, we determined their microbiological safety, chemical composition, minerals and heavy metals, fatty acid and amino acid profile, as well as a sensory profile of the meat. The obtained results showed that the meat from crayfish was microbiologically safe. Crayfish meat has a high nutritional quality, high protein content (18.12%) and a total of 17 detected amino acids, of which essential amino acids constituted 6.96 g/100 g sample. Additionally, the crayfish meat was characterized by high levels of essential polyunsaturated fatty acids (PUFA), particularly n-3 PUFA, at an optimal ratio of n-3/n-6 and with low values of atherogenic and thrombogenic indices. Predominant macrominerals in the meat are K, followed by Na, Ca, P and Mg, whereas the content of microminerals was in the following order: Zn > Cu > Fe > Mn. The concentrations of accumulated toxic metals (Cd, Pb, As and Hg) did not exceed the maximum allowed levels. Sensory analysis confirmed that the meat from spiny-cheek crayfish has the potential to become a new food source of essential nutrients.

## 1. Introduction

The spiny-cheek crayfish *(Faxonius limosus, Rafinesque, 1817)* is one of the most important aquatic invaders in European inland waters. It was introduced to Europe more than a century ago and was recorded in more than 20 European countries. From the end of the 19th century up to now, it has become the most widespread non-indigenous crayfish species in Europe, and was included on a list of Invasive Alien Species (IAS) of universal concern within EU regulations [1,2,3,4,5].

According to the data, spiny-cheek crayfish progressively colonized and increased in population in the Danube River Basin in Hungary, Croatia, Romania and Bulgaria [6,7]. The first record of spiny-cheek crayfish in Serbia was in the Danube River near Apatin in 2002 [8]. Nowadays, this species broadens its range of colonization, both upstream and downstream, along the entire Serbian section of the Danube River and its tributaries [9,10]. According to Hudina et al. [11] the downstream spread rate of *Faxonius limosus* in the Muri River, Croatia, was 18–24.4 km.yr(−1), whereas a rate of 13 km. yr(−1) was recorded in the Danube River, Hungary [12].

Considering the available data and high dispersal rate of these species, it can be assumed that its invasive range in Serbia is larger than it is documented with a tendency for a high degree of expansion in the future [9]. This IAS represents a serious threat to native crayfish (*Astacus astacus* and *Astacus leptodactylus*), which are on the IUCN Red List of Threatened Species [13], as well as being a threat to other aquatic organisms because it feeds on aquatic vegetation, fish eggs and invertebrates, and thus affects biodiversity [13]. The IUCN is working towards raising awareness of the importance of freshwater species, and increasing their representation on the IUCN Red List through assessments of the following freshwater taxonomic groups: decapods (crabs, crayfishes, shrimps); fishes; dragonflies and selected aquatic plants [14].

The countries of the European Union through which the Danube River flows follow and carry out the strategy of preventing the introduction, control or eradication of adopted IAS according to the listed regulations [1]. Some of these countries, such as Croatia, have adopted and have been implementing a national strategy since 2018, i.e., a dedicated law on IAS as a legal basis for the implementation of EU regulations [1]. Prevention refers to the prevention of the spread of new IAS in the territory of Croatia from neighbouring countries, as well as the prevention of the further spread of already present IAS to new areas within the Republic of Croatia. In order to successfully control populations and slow down the spread of IAS, Croatia combines several methods of removal, which implies an integral approach: catching crabs, monitoring the daily catch by recording the sex and size of the individuals and taking care of the caught crabs by handing them over to a registered entity that collects and transports animal by-products. A country that is not a member of the European Union (Serbia) does not harmonize its regulations with EU regulations [1]. To date, there have been no dedicated institutional frameworks for IAS in Serbia. Problems with IAS are solved on a case-by-case basis if there are any at all [15]. As the IAS does not admit borders between countries, insufficient cooperation and communication between countries are one of the main reasons for the failure of this strategy [5].

The spiny-cheek crayfish shows several characteristics such as rapid maturation, a short lifespan, high fecundity and a second mating period, which facilitate its fast population growth, giving spiny-cheek crayfish high invasive potential. Additionally, the negative impact of the spiny-cheek crayfish on the native crayfish populations in Europe is expressed in competition for habitats, in which the invader is more adaptive, and as a carrier of crayfish plague, it is lethal to the European native crayfish, and can destabilize riverbanks and modify other habitats because of its burrowing behavior, causing huge economic damage [16]. Generally, economic damage caused by IAS could cost Europe billions of euros per year and damage costs are continuing to rise [17]. Smart exploitation and processing of crayfish (meat and shell) into new food/feed/polymer products has been sought as a promising solution to reduce the negative impact of spiny-cheek crayfish.

Crayfish meat is recognized as consumable meat with a delicious taste. It has been reported to have high nutritional value with superior biological value through digestibility, use of proteins, high content of essential amino acids and protein efficiency [18]. In general, crayfish meat is characterized by high protein content (18–20%), relatively low-fat content (0.14–1.69%) and favorable fatty acid composition (rich in omega-3 fatty acids). This meat is easily absorbed by the body and has a low energy value of 76 kcal [19,20]. Previous research has revealed that crayfish meat is an excellent source of minerals, particularly calcium, sodium, potassium, copper, zinc, magnesium and iron [21,22]. In comprehensive literature reviews, only a few scientific studies have investigated spiny-cheek crayfish meat [23,24,25]. Thus, there is insufficient data that fully describes the safety, nutritive value and sensorial properties of spiny-cheek crayfish meat, which are a prerequisite for their exploitation and placement as a novel product in the market. In line with the abovementioned and the fact that in recent years the food industry has been increasingly interested in alternative sources of animal material and the development of new product lines, this study aimed to upgrade the research data regarding the safety, technological quality, nutritive properties and sensory value of spiny-cheek crayfish meat as a new generation of food.

## 2. Materials and Methods

### 2.1. Sampling of Spiny-Cheek Crayfish

This study was carried out along the main Danube river course flowing through Stari Slankamen, Inđija, Serbia. Specimens of *F. limosus* were sampled from mid-April to mid-November 2021. Sampling was conducted monthly by the “River Crayfish” association. The sampling procedure involved standard cages used to capture baited crayfish [26]. In accordance with the research [26], the catch methodology performed involves a cage 80 cm long and 25 cm high and wide, in a mesh up to 5 mm. The entrance corresponded to a cone with a length of 15 cm and a diameter of 6 cm on the smallest side (Figure 1); as bait, local fish were used in each.

The traps were set up in appropriate places on the river (depth range of 1–8 m, with slower current speed and proximity to an adequate shelter) for 24 h. During the setting of the traps, they were interconnected by a rope at regular intervals of 4 m, producing a series of 5 traps [26]. For the selected locality, a series of five traps were set up monthly. Trapped specimens of native species were characterized at the sampling site and immediately afterward returned undamaged to their natural habitat. During transport, the captured spiny-cheek crayfish were placed in containers with water taken at the sampling point, equipped with an aeration pump and a cooling system to maximize the health and survival of individuals [27]. Upon transport of samples to the laboratory and measurement of each individual, the low temperature (freezing) method in the thermal shock chamber (−40 °C), which is non-invasive and considered humane [27], was used to kill captured spiny-cheek crayfish individually. This method does not use additional chemicals that could disrupt subsequent chemical analyses of the samples. The samples were deep-frozen and stored at a temperature of −18 °C until further analyses. A sample of edible abdominal muscle tissue collected from each individual crayfish was used for analysis. The average weight of the meat of the cold sample was 34 g.

### 2.2. Microbial Safety

Microbiological safety was determined by enumeration of the total count of bacteria [28] followed by pathogenic bacteria: *Salmonella* spp. [29], beta-glucuronidase-positive *Escherichia coli* [30], coagulase-positive staphylococci [31] and *L. monocytogenes* [32].

### 2.3. Chemical Composition

The moisture and crude ash content were determined according to ISO methods [33,34]. Fat content was determined by Soxhlet extraction with prior acid hydrolysis [35]. Determination of the total phosphorus content expressed as P_2_O_5_ was performed according to ISO methods [36]. Protein content was assessed by the Kjeldahl method [37], using a nitrogen to protein conversion factor of 6.25. The content of carbohydrate of the samples was determined by Omole et al. [38] calculating 100% (% crude protein + % crude fat + % crude ash + % moisture), whereas the gross nutritive value (Kcal/100 g sample) was calculated by multiplying the crude fat (9 Kcal/g), protein (4 Kcal/g) and carbohydrate (4 Kcal/g) as described in the regulations (EU) [39].

### 2.4. Fatty Acid Composition

The lipids from abdominal crayfish meat were extracted with chloroform:methanol (2:1, vol/vol) using the method described by Folch et al. [40]. Fatty acid methyl esters were then prepared from the extracted lipids by transmethylation that uses 14% (*w*/*w*) boron tri-fluoride in methanol solution. The methyl esters were analysed with Agilent 7890A gas chromatography (Agilent Technologies, Santa Clara, CA, USA) equipped with a fused silica capillary column (Supelco SP-2560 Capillary GC Column 100 m × 0.25 mm, d = 0.20 μm) (Supelco, Bellefonte, PA, USA) and a flame ionization detector (FID). Helium was used as a carrier gas, whereas the detector and injector temperatures were chosen as 240 °C and 250 °C, respectively. The fatty acid methyl esters were identified by comparing the retention times with those of a mixture of external standard methyl esters (Supelco 37 Component FAME Mix, Sigma-Aldrich, St. Louis, MI, USA). The percentage of composition of individual fatty acids was calculated by using the method of internal normalization. The atherogenic (AI) and thrombogenic (TI) indices were calculated using the Ulbricht and Southgate [41] equations.

### 2.5. Amino Acid Composition

The amino acids profile of crayfish abdominal samples were estimated by ion exchange chromatography using an automatic amino acid analyzer Biochrom 30+ (Biochrom, Cambridge, UK), according to Spackman et al. [42]. The Biochrom 30+ analyzer provides accurate quantitative analysis of amino acid mixtures and the technique was based on amino acid separation using strong cation exchange chromatography, followed by the ninhydrin colour reaction and photometric detection at 570 nm, except for proline, which was detected at 440 nm. Samples were previously hydrolysed in 6M HCl (Merck, Germany) at 110 °C for 24 h. After hydrolysis, the samples were cooled to room temperature (20 °C) and dissolved in 25 mL of loading buffer (pH 2.2) (Biochrom, Cambridge, UK). Samples were then filtered through a 0.22 μm pore size PTFE filter and the filter residue was transferred into a vial (Agilent Technologies, Santa Clara, CA, USA). The filtrate was then stored in a refrigerator and analyzed. The amino acid peaks were identified by comparison of retention times with the retention times of standard amino acids purchased from Sigma Aldrich (Amino Acid Standard Solution (Sigma-Aldrich, St. Louis, MI, USA)). The results were expressed as mass of amino acid (g) in 100 g of sample.

### 2.6. Macro and Micro Elements and Heavy Metals

The content of macro (K, Ca, Mg, Na, P) and micro elements (Fe, Mn, Cu) was determined according to the ISO method [43] based on atomic absorption spectrometry using a flame atomic absorption spectrometer (Varian SPECTRA AA-10, Varian Techtron Pty Limited, Mulgrave, Victoria, Australia) equipped with a flame furnace and operated with an air acetylene flame after mineralization by dry aching. Heavy metal analysis (Pb, Hg, Cd, and As) was performed by atomic absorption spectrometry after dry ashing according to the EN method [44]. An Advanced Mercury Analyser AMA 254 (Altec, Prague, Czech Republic) was used for Hg analysis in the crayfish meat samples. This instrument is principally based on the combustion of the sample in a combustion chamber flooded with oxygen [45]. Results were expressed as mg of mineral content on 100 g of meat samples.

### 2.7. Sensory Analysis

Descriptive sensory analysis was carried out by the sensory panel consisting of thirty professional culinary chefs (7 females and 23 males of ages between 30 and 55 years) selected for their expertise (at least three years of experience as chef or culinary instructor), knowledge and availability to participate in the study. All participants have previous experience in detecting, describing and discriminating sensory attributes in regular university coursework. Culinary professionals work with their senses and may thus be considered as traditional sensory experts [46]. Ninety (90) abdominal muscle crayfish were heat-treated for sensory analysis. The samples were heat-treated by poaching, sautéing or frying in shallow fat. Poaching was done in slightly acidified water with acetic acid at a temperature of 80 °C for 2 min. Sautéing was done in seasoned butter at a temperature of 71 °C for 3 min. Shallow-fat frying took place in refined grape seed oil at a temperature of 165 °C for 2 min. In a prior testing session, panelists agreed about 32 descriptors related to the crayfish meat’s appearance (white color nuance, meat shines, dots visibility), odor (overall odor intensity, algae odor, fresh fish odor, cooked fish odor, butter odor, oil odor, shrimp odor, mud odor), taste (sweet, salty, bitter, sour, umami), flavor (algae flavor, fresh fish flavor, cooked fish flavor, butter flavor, oil flavor, shrimp flavor, caramel flavor, flavor persistence) and texture (hardness, elasticity, gumminess, softness, juiciness, chewiness, greasiness), which were then used for sensory profiling. The intensity of perceived sensory descriptors was evaluated on a scale from 0 (not perceivable) to 7 (strongly perceivable) in steps of 1. The samples (two pieces of crayfish meat per sample) were delivered one by one in odorless plastic containers (served within 5 min after preparation) codified with three digits random numbers together with a cup of water for palate cleansing. The evaluation was performed in individual sensory booths equipped with appropriate ventilation, lighting and a controlled temperature. All participants received written information about the study and provided written, informed consent to participate.

### 2.8. Statistical Analysis

Results represent a mean value ± standard deviation of three individual measurements. Statistical analysis of the data and significant differences at the significant level 0.05 for all variables were analyzed by the one-way ANOVA procedure, followed by Tukey’s HSD multiple comparison tests using software package XLSTAT 2018.7. (Addinsoft, New York, NY, USA).

### 2.9. Ethical Statement

The ethical statement was obtained from the Ethics Committee for Protection and Welfare of Experimental Animals at the University of Novi Sad, Serbia (no. 04-81/94, EK: I-2020-06-1). The ethical statement was stated based on the National Regulations (Annex 4 of the “Official Gazette of RS”, No. 39/10), which clearly indicates that ethics approval for research involving invertebrate animals is not required. For the purpose of sensory evaluation of the samples of cooked meat, human participants were involved as members of a trained sensory panel. The ethical statement was obtained from the Ethical Committee for Clinical Trials from the Faculty of Medicine, University of Novi Sad, no. 01-39/239/1.

## 3. Results

The safety assessment (cfu/g) of spiny-cheek crayfish meat is presented in Table 1. Examined pathogenic bacteria *Salmonella* spp., beta-glucuronidase-positive *Escherichia coli*, coagulase-positive staphylococci and *L. monocytogenes* were below the detection limit of <10 or <100/cfu/g. The total count of bacteria was generally low and was only 10^3^ cfu/g in the tested sample.

The data presented in Table 2 show the chemical composition and nutritive value of raw meat from the spiny-cheek crayfish. Obtained results showed that raw crayfish meat has a high proportion of water ~80%, that is, approximately 20% of dry matter. Additionally, crayfish meat had a gross nutritive value between 311.18 and 318.60 kcal/100 g. The protein content in crayfish accounted for 18.12%, whereas the content of ash, fats and carbohydrates were very low and accounted for 1.37, 0.25 and 0.10%, respectively. Of the total fats, total polyunsaturated fats (0.11%) were the highest contributor in the spiny-cheek crayfish meat. No significant differences were found between the shares of saturated and monounsaturated fats in the meat. Of the polyunsaturated fats, omega-3 fats were the present in the largest share (0.07%), whereas omega-6 fats were distributed in the amount of 0.04%.

The fatty acid content presented in grams per 100 g of total fatty acids is shown in Table 3. The analyzed fatty acid composition of crayfish meat was dominated by polyunsaturated fatty acids (PUFA), which comprised 45.76%, whereas saturated (SFA) and monounsaturated (MUFA) fatty acids were distributed in the amount of 27.34 and 26.90%, respectively. Among SFA, the most dominant was palmitic acid (~18%), followed by stearic acid (~7%). Myristic, heptadecanoic and pentadecanoic acids were presented in lower amounts. In crayfish meat, oleic acid was the major MUFA, whereas palmitoleic, pentadecenoic and heptadecenoic acid were less abundant. In the n-3 PUFA group, the most abundant fatty acids were eicosapentaenoic acid (EPA) and docosahexaenoic acid (DHA) with an amount of around 25 and 4%, respectively. Alfa-linoleci acid was presented in the amount of 0.55%. Arachidonic and linoleic acid were the major n-6 PUFA observed in the crayfish meat (9.18%), followed by linoleic (5.29%) and eicosadienoic acid (1.95%). The ratio of n-3/n-6 in the crayfish meat was 1.79. Furthermore, AI and TI were found to be 0.40 and 0.21, respectively, in spiny-cheek crayfish.

The content of minerals and heavy metals in the crayfish meat are shown in Table 3. The results showed that the predominant macromineral in the meat was K (440.33 mg/100 g), followed by Na, Ca, P and Mg (99.67, 52.17, 50.33 and 25.60 mg/100 g, respectively), whereas the content of microminerals was in this following order: Zn > Cu > Fe > Mn (2.53, 2.09, 0.98 and 0.12 mg/100 g, respectively). As shown in Table 3, the concentration of toxic metals Pb, Cd and As in muscle meat was lower than the detection limit of the instrument or analytical method. The low concentration of Hg was found in samples of 0.025 mg/100 g and the highest composition did not exceed the concentration of 0.027 mg/100 g.

In the present study, a total of 17 amino acids were detected in the spiny-cheek crayfish and the results are shown in Table 3. The content of total amino acids in the meat was 17.62 g/100 g of the samples and the values ranged from 0.46 to 2.46 g/100 g of the samples. Amino acids, glutamic acid (Glu), arginine (Arg), aspartic acid (Asp), Lysine (Lys) and Leucine (Leu) were the most abundant. The level of essential amino acids (EAAs) was lower than the level of nonessential amino acids (NEEAs) and accounted for 39.50% of the total amino acids. For EAAs, Lys was detected in the highest amount in the sample, whereas Glu and Arg were the most abundant among non-essential amino acids (NEAAs).

A spider plot was created to observe differences in the sensory profiles of the crayfish samples prepared with three different preparation procedures (Figure 2). All observed sensory properties of the samples of crayfish meat were from barely to moderately noticeable. Frying with butter and roasting crayfish meat significantly (*p* < 0.05) increased butter and oil odor and flavour, respectively, whereas at the same time the perception of fresh fish odor and flavour, cooked fish odor and flavour significantly (*p* < 0.05) decreased and was the most noticeable in Sample 2. Moreover, less desirable sensory properties such as mud odour and algae flavour was the most perceived in Sample 2 as well. Cooking procedure significantly (*p* < 0.05) affected the appearance and taste properties of crayfish meat. Sample 1 prepared with frying in butter was the darkest and possessed the most noticeable sweet taste in comparison to the other samples. Differences in terms of analysed textural properties were not significant (*p* > 0.05) between the samples.

## 4. Discussion

The purpose of this study is to provide information on the safety and characteristics of spiny-cheek crayfish meat to help formulate a comprehensive control plan in other river crayfish habitats. The results shown in Table 1 indicate that the microbiological safety of tested crayfish meat samples was consistent with parameters prescribed by the Commission Regulation (EC) No. 2073/2005 [47] on microbiological criteria for foodstuffs and its amendment Regulation (EC) No. 1441/2007 [48]. In general, the presence of microorganisms on aquatic animals depends on various factors including season, water quality, temperature and locality [49]. According to the ICMSF, most aquatic animals at the time of harvesting have bacterial counts in the range from 102 to 105 cfu/g. In addition, in research conducted by Lalitha et al. [50] and Leitào and Rios [51], the initial total bacterial count of mainly freshwater prawn ranged from 102 to 106 cfu/g, which is in line with our observation.

The present research evidenced that crayfish meat was mainly constituted of proteins, whereas, the content of lipids was insignificant (Table 2). This observation is in agreement with Śmeitana et al. [52] for crayfish *F. limosus* caught in Polish waters. However, El-Sherif et al. [53] and Papachristou et al. [54] analyzed Nile River crayfish (*Procambarus clarkia*) and found a significantly lower amount of protein and higher amount of fat than those in our study.Furthermore, El-Kholie [55] reported an eight times higher protein content, slightly higher dry matter content and gross nutritive value in the meat of Nile River crayfish. The differences in chemical composition could be caused by taxonomic differences in species, life environment riverine habitats, season of year and sex [52]. The protein content in crayfish meat is similar to that in the meat originating from beef, pork, poultry, marine and freshwater fish, crab and shrimp. However, the content of fat in crayfish is considerably lower than that in the meat of game (wild boar, deer, moose) and farmed animals, but comparable with that in the meat of other crustaceans (oysters, crab, shrimp) [25,52]. According to El-Sherif et al. [53], despite the high protein content and gross nutritional values that meet the recommended level and provide enough calories according to the recommendations for the daily activities of FAO [56], the use of crayfish meat is still neglected by consumers. Furthermore, the low fat content and energetic value of this meat suggest its potential use in hypocaloric human diets. This may be because of the small edible part which represents about 18% [53].

The fatty acid profile is a valuable indicator for the assessment of the quality of edible aquatic species. The results presented in Table 3 argue that crayfish meat has a lower content of SFA an MUFA but a higher level of PUFA, of which the most dominant were n-3 PUFA, especially EPA and DHA, indicating a higher nutritional quality of meat. Crayfish meat had a more favorable profile of FA in comparison to meats such as beef, pork, chicken and lamb [54]. Crustaceans generally have balanced fatty acid content as they have a significant amount of essential n-3 and n-6 PUFA, which are necessary for human health. These fatty acids cannot be synthetized by the human body and therefore must be derived from diet. The fatty acid profile in crayfish meat was comparable to that reported by Śmietana [52]. On the other hand, Stanek et al. [22] observed lower n-3 PUFA but higher n-6 PUFA in the crayfish meat found in our study. Harlioğlu et al. [57] reported a higher content of DHA, but a lower content of linolenic and arachidonic acid in wild caught narrow-clawed crayfish (*Astacus leptodactylus)*. The content of EPA was higher than that observed in some species of marine crustaceans [57,58]. Fatty acid content in freshwater and marine crustaceans readily depends on the fatty acid composition of the diet and the content of the dietary lipid source. Environmental factors such as temperature, life cycle, moult stage and reproductive performance can modify the fatty acid composition in the muscle as well [57]. From the point of nutritional quality, the consumption of higher levels of n-3 PUFA is preferable over the dietary inclusion of higher levels of n-6 PUFA, which are involved in the inflammatory process. Whereas, on the other hand, n-3 PUFA are associated with a reduced risk of cardiovascular and liver diseases, arthritis, thrombosis, inflammation, autoimmune disorders and certain types of cancer [59]. DHA that is highly abundant in crayfish meat is vital for the normal brain function and development. The ratio of n-3/n-6 in crayfish meat was 1.79 and was in compliance with the results previously reported for spiny-cheek crayfish [22,52]. The World Health Organization recommended that the ratio of n-3/n-6 should be 1-2/1, which indicates that crayfish meat represents an excellent food source for humans. Atherogenic (AI) and thrombogenic indices (TI) are indicators of the lipid nutritional quality of and their potential impact on the development of coronary disease. Generally, AI and TI should be kept at low levels (<1.0 for AI and <0.5 for TI) [24]. The observed AI and TI of the crayfish meat were lower than those for other meats such as beef, lamb, pork, chicken and rabbit [23]. The authors revised the sentence: High level of n-3 PUFA, low n-3/n-6 ratio, AI and TI indicated that the meat of spiny crayfish could be considered as an alternative source of essential fatty acids and that regular consumption of crayfish meat has great potential for protecting against coronary disease [22,24].

As observed in Table 4, crayfish meat contains high contents of K, Na, Zn and Fe, which have many biological roles and which are necessary for the improvement of healthy body functions, metabolic processes and normal cell function as well as for the storage and generation of energy in the body [60,61]. Zn, Mn and Cu are considered as elements essential for life; however, they may have a toxic effect when accumulated in excessive amounts. It was reported that crayfish can accumulate minerals from sediment, water or even through diet [60,61]. Śmietana et al. [24] reported a similar content of macrominerals, with the exception of P, for which the content was five times higher (274.6 mg/100 g) than that found in our study, and a significantly lower content of micromineral Cu (0.38 mg/100 g). A similar content of Cu but a higher content of Zn and Mn in the meat of crayfish was observed in the previous studies [60,61]. The content of bio-accumulated minerals in crayfish meat depends on the species, environment and season of the crayfish acquisition [52]. Crayfish are considered good bioindicators for monitoring aquatic environmental pollution [60,61]. Taking into consideration that the spiny-cheek crayfish lives in benthic habitats with constant contact with bottom sediments, there is a high risk of transferring toxic metals from sediment and water into crayfish and their bioabsorption and bioaccumulation in the meat, which poses a serious risk when consuming the contaminated meat. Heavy metals present a huge threat to human health as they have mutagenic and carcinogenic effects, and can cause reproductive dysfunction, cognitive disorders and kidney failure [61]. Therefore, determination of the non-essential toxic metal content in crayfish meat is a very important step in food safety and risk assessments [61]. As presented, the concentration of the toxic metals Cd, Pb, As and Hg in crayfish meat were below the maximum limits established by the European Commission [62]. According to the aforementioned regulations, the maximum concentration of Cd, Hg and Pb in the muscle of crustaceans should not exceed 0.50 mg/kg wet weight. As shown, the presence of Hg was determined from the metal pollutants, but its content was significantly below the maximum limits during the entire capturing season. Various factors can influence the level of heavy metals in crayfish: species, location, size, diet, type of tissue, as well as the period of exposure to environmental factors [60]. In the research by Protasowicki et al. [61], it was observed that the entire body of the crayfish had a significantly higher concentration of heavy metals than that in the abdominal muscles of the crayfish. They also determined that the heavy metal concentration in crayfish highly depends on the environment where the crayfish were collected. Suárez-Serrano et al. [63] observed a high level of mercury (3.5 mg/kg) exceeding the maximum level in the muscle of crayfish collected near sediment waste. Stanek et al. [60] also reported that the content of toxic Pb in the muscle of crayfish was higher than the maximum allowed levels for fish and crayfish intended for human consumption. On the other hand, other studies reported that the level of heavy metals in the abdominal meat of crayfish did not exceed the permissible limits [52]. In our study, a low level of accumulated heavy metals in the meat indicates that it is toxicologically safe for the health of potential consumers to eat meat from the crayfish captured in the studied area.

In this research, 17 amino acids were detected in the crayfish meat (Table 5). Tryptophan was lost during the acid hydrolysis of the meat sample and thus was not quantified, but its concentration in crustaceans and sea food is usually neligable [54].The nutritional quality of protein in the diet is particularly related to the concentration of essential amino acids, which cannot be synthesized in the human body and therefore must be acquired through nutrition [54]. Around 40% of total AA were EAA, of which Lys and Leu were the most dominant. The results are consistent with a previous study by Śmietana et al. [24] who discovered a high amount of Lys in crayfish (1.41%), which is an essential amino acid for humans and animals, and leucine (1.31%) and the most dominant nonessential amino acids, Glu and Arg, with a percentage of 2.96 and 1.94%, respectively. The meat of edible crab *Cancer pagurus* was shown to have a lower concentration of Lys (0.77–1.06 g/100 g) but a higher concentration of glycine (Gly) (1.23–1.36 g/100 g), depending on the sex and season [64]. Higher concentrations of histidine (His) (1.98 g/100 g), Glu (3.16 g/100 g), Asp (2.34 g/100 g) and serine (Ser) (1.05 g/100 g) were observed in the meat from red crayfish from the River Nile [55]. Wang et al. [65] reported higher concentrations of some EAASs, such as Lys, valine (Val), threonine (Thr) and phenylalanine (Phe) in the muscle of wild-caught mitten crabs caught from different river basins.

In order to complete the research after the confirmation of food safety of River spiny-cheek crayfish meat, a sensory analysis was carried out, which provides useful information for understanding the influence of different preparation procedures on the sensory properties of crayfish meat, and in what way shortcomings associated with this type of food can be overcome by applying appropriate preparation procedures. The overall low intensity scores are in accordance with previously reported studies whereby fresh seafood is generally characterized by mild flavors [66,67]. The lower acceptability of Sample 2—crayfish meat prepared with pouching—is probably due to the perceptible odor and flavor on algae, which are generally considered as unpleasant [67], whereas on the other hand, the higher acceptability of Sample 1—crayfish meat prepared with sautéing in butter—is probably due to the dominant odor and flavor of butter, the flavor of caramel and sweet taste.

## 5. Conclusions

The increasing prevalence of spiny-cheek crayfish in the Danube river basin emerged as a good opportunity for its exploration in food processing industry. Abdominal meat from the spiny-cheek crayfish captured in the Danube River was shown to be a rich source of proteins, essential amino acids and fatty acids, as well as some minerals. Meat had a high level of polyunsaturated fatty acids, particularly n-3 fatty acids, a favourable ratio of n-3/n-6 fatty acids, low values of atherogenic and trombogenic indices as well as acceptable sensory properties. The health risk assessment attested that meat from the crayfish captured in the studied area is microbiologically and toxicologically safe concerning heavy metals. The results clearly demonstrate that the meat from crayfish has a high nutritional value and has a good potential to serve as a new food source. Since crayfish can accumulate heavy metals from sediments and water, further research would include investigation of crayfish captured from other Danube streams to get a more realistic picture of the safety and nutritional properties of crayfish meat.

## Figures and Tables

**Figure 1 ijerph-19-16819-f001:**
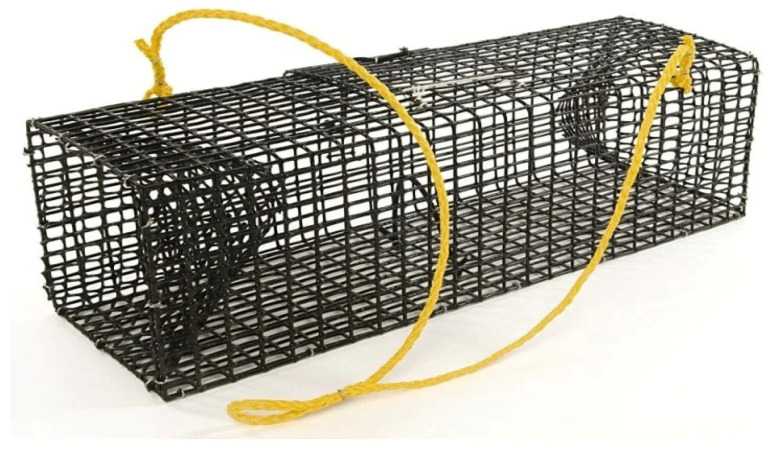
Crayfish trap.

**Figure 2 ijerph-19-16819-f002:**
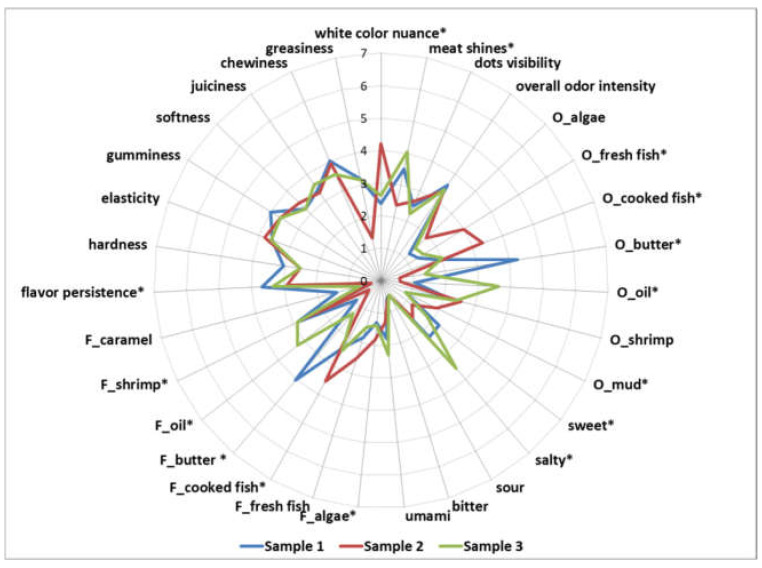
Spider plot of sensory profiles of crayfish meat samples prepared in three different cooking procedures (Sample 1—crayfish meat prepared with sautéing on butter, Sample 2—crayfish meat prepared with poaching; Sample 3—crayfish meat prepared with shallow-fat frying; * next to the descriptor name indicates statistical difference of means determined by Tukey HSD test at *p* < 0.05) (O_ in the name of descriptor indicates odor attribute, whereas F_ in the name of descriptor indicates flavor attribute).

**Table 1 ijerph-19-16819-t001:** Microbiological profile (cfu/g meat) of spiny-cheek crayfish.

Microorganisms	Mean Average (cfu/g)	Range
Total bacterial count	1000 ± 0.11	500–1500
*E.coli*	<10	<10
*L. monocytogenes*	<100	<100
*Salmonella* spp.	not detected in 25 g	not detected in 25 g
*Coagulase positive Staphylococcus*	<100	<100

**Table 2 ijerph-19-16819-t002:** Nutritive value of spiny-cheek crayfish.

Nutrients (% of Wet Weight)	Quantity (Mean Average)	Range
Gross nutritive value (kJ/100 g)	75.15 ± 0.73	74.34–76.11
Gross nutritive value (Kcal/100 g)	314.58 ± 0.70	311.18–318.60
Moisture (%)	80.15 ± 0.22	79.84–80.34
Ash (%)	1.37 ± 0.05	1.30–1.42
Proteins (%)	18.12 ± 0.21	17.90–18.40
Carbohydrates (%)	0.10 ± 0.01	0.09–0.11
Fats (%)	0.25 ± 0.02	0.23–0.27
Saturated fats (%)	0.07	0.05–0.08
Monounsaturated fats (%)	0.07	0.06–0.07
Polyunsaturated fats (%)	0.11	0.10–0.12
Omega-3 fats (%)	0.07	0.06–0.08
Omega-6 fats (%)	0.04	0.03–0.05

**Table 3 ijerph-19-16819-t003:** Fatty acid composition, atherogenic (AI) and thrombogenic (TI) indices of spiny-cheek crayfish meat.

Fatty Acid	% of Total Fatty Acids (Mean Average)	Range
^1^ SFA	27.34 ± 0.48	24.61 ± 0.25–28.66 ± 0.32
Myristic acid (C14:0)	1.09 ± 0.10	0.75–1.32
Pentadecanoic acid (C15:0)	0.70 ± 0.90	0.59–0.92
Palmitic acid (C16:0)	17.78 ± 0.15	16.60–18.26
Heptadecanoic acid (C17:0)	0.90 ± 0.07	0.65–1.06
Stearic acid (C18:0)	6.87 ± 0.11	6.02–7.10
^2^ MUFA	26.90 ± 0.36	26.35 ± 0.20–27.34 ± 0.42
Pentadecenoic acid (C15:1)	0.33 ± 0.18	0.20–0.65
Palmitoleic (C16:1)	4.42 ± 0.22	3.59–5.03
Heptadecenoic acid (C17:1)	0.33 ± 0.02	0.19–0.42
Oleic acid (C18:1 n-9)	20.60 ± 0.16	20.16–20.99
Eicosenoic acid (C20:1 n-9)	1.22 ± 0.02	1.12–1.35
PUFA	45.76 ± 0.84	44.44 ± 0.55–46.50 ± 0.68
^3^ n-3 PUFA	29.34 ± 0.39	27.75 ± 0.18–30.00 ± 0.25
alfa-Linolenic aci(C18:3 n-3) (ALA)	0.55 ± 0.15	0.31–0.70
Eicosappentaenoic acid (C20:5 n-3) (EPA)	24.94 ± 0.33	24.12–25.71
Docosahexaenoic acid (C22:6 n-3) (DHA)	3.85 ± 0.24	3.15–4.28
^4^ n-6 PUFA	16.42 ± 0.14	14.44 ± 0.10–17.10 ± 0.08
Linoleic acid (C18:2 n-6) (LA)	5.29 ± 0.20	4.51–6.39
Eicosadienoic acid (C20:2 n-6)	1.95 ± 0.05	1.85–2.02
Arachidonic acid (C20:4 n-6) (AA)	9.18 ± 0.90	8.08–10.20
n3/n6 ratio	1.79 ± 0.01	1.92–1.75
AI	0.40 ± 0.01	0.39–0.40
TI	0.21 ± 0.01	0.20–0.21

^1^ SFA (saturated fatty acids):C14:0 + C15:0 + C16:0 + C17:0 + C18:0; ^2^ MUFA (monounsaturated fatty acids): C15:1 + C16:1 + C17:1 + C18:1 n-9 + C20:1 n-9; ^3^ PUFA (polyunsaturated fatty acids) n-3: C18:3 n-3 + C20:5 n-3 + C22:6 n-3; ^4^ n-6 PUFA: C18:2 n-6 + C20:4 n-6 + C20:2 n-6.

**Table 4 ijerph-19-16819-t004:** Minerals and heavy metals contents of spiny-cheek crayfish meat.

Minerals and Heavy Metals	Quantity (mg/100 g)	Range
Macrominerals		
K	440.33 ± 89.67	428.50–450.20
Ca	52.17 ± 6.24	51.50–53.01
Mg	25.60 ± 9.24	24.50–26.80
Na	99.67 ± 41.10	95.01–105.00
P	50.33 ± 35.20	46.50–55.00
Microminerals		
Fe	0.98 ± 0.54	0.92–1.05
Mn	0.12 ± 0.04	0.11–0.13
Cu	2.09 ± 0.09	2.07–2.10
Zn	2.53 ± 0.33	2.49–2.59
Heavy metals		
Hg	0.025 ± 0.04	0.019–0.029
Pb	<0.1	<0.1
Cd	<0.02	<0.02
As	<0.1	<0.1

**Table 5 ijerph-19-16819-t005:** Amino Acids profile of spiny-cheek crayfish meat in g/100 g of sample.

Amino Acids	Quantity (g/100 g)	Range
His	0.59 ± 0.04	0.55–0.62
Ile	0.78 ± 0.10	0.72–0.80
Leu	1.37 ± 0.06	1.30–1.40
Met	0.46 ± 0.05	0.40–0.50
Phe	0.71 ± 0.07	0.65–0.78
Thr	0.80 ± 0.01	0.72–0.84
Val	0.74 ± 0.04	0.70–0.85
Lys	1.51 ± 0.03	1.49–1.54
Total EAAs	6.96 ± 0.40	6.53–7.33
Glu	2.46 ± 0.04	2.42–2.50
Asp	1.58 ± 0.02	1.50–1.65
Pro	0.94 ± 0.05	0.89–0.98
Ala	0.85 ± 0.06	0.78–0.90
Arg	1.98 ± 0.11	1.88–2.10
Gly	0.86 ± 0.08	0.82–0.90
Ser	0.66 ± 0.04	0.62–0.69
Tyr	0.82 ± 0.11	0.82–1.02
Cys	0.11 ± 0.05	0.10–0.15
Total NEAAs	10.26 ± 0.65	9.83–10.89
Total AAs	17.22 ± 1.05	16.36–18.22

EAAs—essential amino acids, NEAAs—non-essential amino acids, TAAs—total amino acids.

## Data Availability

Data are contained within the article.

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
