# Peer review of "Invasive Crayfish Faxonius limosus: Meat Safety, Nutritional Quality and Sensory Profile"

_ijerph, 2022, doi:10.3390/ijerph192416819_

Round 1

Author Response

Dear Review 1,

We thank the reviewer for their careful reading of the manuscript and their constructive remarks. We have taken the comments on boards to improve and clarify the manuscript. We outline below our responses (Word file-attach). All remarks are accepted and the paper is changed according to these comments. We have written bullet points of our major changes to the manuscript, including a "track changes" document.

Best regards,

 PhD Jasmina lazarević- author

Reviewer 2 Report

I received a publication for review on "Invasive crayfish Faxonius limosus: Meat Safety, Nutritive Quality and Sensory Profile"; ijerph-2064252

The subject matter is interesting and the publication contains a lot of important information. However, I believe that it needs some improvements.

L: 202 - is deleted by mistake "s" in the word samples

L: 207 is "Table 1" should be "Table 2"

L: 223 is "Table 2" should be "Table 3"

L: 246 is "Table 3" should be "Table 4"

L: 250 is "Table 3" should be "Table 4"

L: 256 is "Table 3" should be "Table 4". In addition, there are no spaces after the word "Table"

In the Table: " Minerals and heavy metals of spiny-cheek crayfish meat", the full names of the elements should be deleted - the abbreviations are obvious and fully sufficient

L: 261: is "Table 3" should be "Table 4"

L: 270 In the description of the Table "s" in the word samples is deleted

Moreover, in the Table "Amino Acids profile of spiny-cheek crayfish meat in g/100g of samples" - amino acid names should be presented only as abbreviations

 L: 312 is "Smietana et al." it should be "Śmietana et al."

L: 337 is "Śmeitana" should be "Śmietana et al."

L: 606 Item No. 58 ie Stanek et al. (2022) must be removed as it has not yet been published. It is undergoing substantive evaluation. Therefore, all text related to this reference should also be removed.

I believe the publication will be improved according to the guidelines and consequently published in IJERPH.

Author Response

Dear Review 2,

We thank the reviewer for their careful reading of the manuscript and their constructive remarks. We have taken the comments on boards to improve and clarify the manuscript. We outline below our responses (Word file-attach). All remarks are accepted and the paper is changed according to these comments. We have written bullet points of our major changes to the manuscript, including a "track changes" document.

Best regards,

 PhD Jasmina lazarević- author

Reviewer 3 Report

This manuscript evaluated the safety parameters, nutritional value, and sensory profile of the meat from spiny-cheek crayfish, providing evidence for the eating of the aquatic invaders. However, the manuscript could be improved for publication.

1. The distribution and amount of spiny-cheek crayfish should be mentioned in the Introduction.

2. I suggest that the authors determine some parameters of the digestibility of spiny-cheek crayfish meat.

3. As the unit of the microbiological profile was log CFU/g meat, the data in Table were wrong. Besides, the authors wrote the unit in the wrong form of ‘long CFU/g’.

Author Response

Dear Review 3,

We thank the reviewer for their careful reading of the manuscript and their constructive remarks. We have taken the comments on boards to improve and clarify the manuscript. We outline below our responses (Word file-attach). All remarks are accepted and the paper is changed according to these comments. We have written bullet points of our major changes to the manuscript, including a "track changes" document.

Best regards,

 PhD Jasmina lazarević- author

Round 2

Reviewer 3 Report

No comments.